# A gene horizontally transferred from bacteria protects arthropods from host plant cyanide poisoning

**Nicky Wybouw[1], Wannes Dermauw[1], Luc Tirry[1], Christian Stevens[2], Miodrag Grbić[3,4], René Feyereisen[5], Thomas Van Leeuwen[1,6]***

[1]Laboratory of Agrozoology, Department of Crop Protection, Faculty of Bioscience Engineering, Ghent University, Ghent, Belgium; [2]SynBioC Research Group, Department of Sustainable Organic Chemistry and Technology, Faculty of Bioscience Engineering, Ghent University, Ghent, Belgium; [3]Department of Biology, University of Western Ontario, London, Canada; [4]Instituto de Ciencias de la Vid y el Vino, Logroño, Spain; [5]Institut National de la Recherche Agronomique, Centre National de la Recherche Scientifique and Université de Nice Sophia Antipolis, Nice, France; [6]Institute for Biodiversity and Ecosystem Dynamics, University of Amsterdam, Amsterdam, Netherlands

**Abstract** Cyanogenic glucosides are among the most widespread defense chemicals of plants. Upon plant tissue disruption, these glucosides are hydrolyzed to a reactive hydroxynitrile that releases toxic hydrogen cyanide (HCN). Yet many mite and lepidopteran species can thrive on plants defended by cyanogenic glucosides. The nature of the enzyme known to detoxify HCN to β-cyanoalanine in arthropods has remained enigmatic. Here we identify this enzyme by transcriptome analysis and functional expression. Phylogenetic analysis showed that the gene is a member of the cysteine synthase family horizontally transferred from bacteria to phytophagous mites and Lepidoptera. The recombinant mite enzyme had both β-cyanoalanine synthase and cysteine synthase activity but enzyme kinetics showed that cyanide detoxification activity was strongly favored. Our results therefore suggest that an ancient horizontal transfer of a gene originally involved in sulfur amino acid biosynthesis in bacteria was co-opted by herbivorous arthropods to detoxify plant produced cyanide.

*For correspondence: thomas.vanleeuwen@ugent.be

**Reviewing editor**: Joerg Bohlmann, University of British Columbia, Canada

## Introduction

Plants have developed a remarkable diversity of chemical defenses to deter herbivores from feeding. Cyanogenesis is one of the most ancient and widespread of these defenses, and more than 2500 plant species are known to synthesize cyanogenic glucosides and cyanolipids as phytoanticipins (constitutive defense compounds present in the plant prior to herbivore attack). Upon tissue disruption by herbivore feeding, cyanogenic glucosides are degraded by the plant β-glucosidases and α-hydroxynitrile lyases, which results in the release of toxic hydrogen cyanide (HCN) and other toxic products such as their aglycones (*Gleadow and Woodrow, 2002*; *Poulton, 1990*; *Spencer, 1988*; *Zagrobelny et al., 2004*; *Figure 1*). The released cyanide is a potent inhibitor of the mitochondrial respiratory chain and has a disruptive effect on various metabolic pathways (*Solomonson, 1981*), thus providing a broad defense against generalist herbivores.

In their arms race with plants, arthropods have evolved several mechanisms to overcome plant cyanogenesis. A well-documented case of co-evolution is found in insect lepidopteran specialists that sequester the ingested cyanogenic glucosides in their own defense against predators. Remarkably, cyanogenic compounds have become so crucial for some species such as burnet moths (Zygaenidae)

**eLife digest** Hydrogen cyanide is a poison that is deadly for most forms of life. Also known as prussic acid, it has killed countless humans throughout history in accidents and during the Holocaust. Hydrogen cyanide is also used by plants to defend themselves against insects and other herbivorous animals.

Many plants produce chemicals called cyanogenic glycosides that can be converted into hydrogen cyanide when the plant is eaten. This is an ancient and efficient defense against all sorts of herbivores, including humans. For instance, cassava is a key source of food in sub-Saharan Africa and South America, but it contains cyanogenic glucosides and is highly toxic if eaten in unprocessed form. However, some insects and mites can thrive on cyanogenic plants, often to the extent of becoming pests on these plants.

Certain moths, such as burnet moths, have gone further and now depend on cyanogenic glucosides for their own defenses against predators such as birds. How these mites and insects are capable of fending off cyanide toxicity has long remained a mystery.

Now Wybouw et al. have identified a mite enzyme that detoxifies hydrogen cyanide to produce a compound called beta-cyanoalanine. Remarkably, the DNA that encodes this enzyme did not evolve in animals but originally belonged to a bacterium. Wybouw et al. show that the gene was transferred to the genome of the spider mite *Tetranychus urticae* perhaps a few hundred million years ago. An equivalent gene was also found in moths and butterflies, which explains why these insects can thrive on plants that produce hydrogen cyanide.

This lateral gene transfer from bacteria to animals is a remarkable coalition of two kingdoms against another, and illustrates a new aspect of the chemical warfare between plants and animals. This study also increases our awareness of the importance of laterally transferred genes in the genomes of higher organisms.

that they not only sequester, but also synthesize these compounds de novo by convergent evolution of the biosynthetic pathways (*Jensen et al., 2011*). Next to sequestration, other mechanisms have evolved to cope with the toxic effects of HCN, such as avoiding the ingestion of cyanogenic compounds or detoxifying HCN upon plant release (*Despres et al., 2007*). In animals, HCN is thought to be detoxified by two main pathways. The enzyme rhodanese converts cyanide into thiocyanate, but this biochemical reaction is not very common and thought to be inefficient (*Beesley et al., 1985*; *Davis and Nahrstedt, 1985*; *Long and Brattsten, 1982*). Alternatively, the conversion of HCN and cysteine into β-cyanoalanine and sulfide has been suggested as the main detoxification pathway in arthropods (*Figure 1*). This is supported by several biochemical surveys showing a correlation between β-cyanoalanine synthesis and HCN exposure in lepidopteran species tolerant to HCN (*Meyers and Ahmad, 1991*; *Stauber et al., 2012*). However, the enzyme that catalyzes this crucial reaction in arthropods has not been identified to date. The conversion of HCN into β-cyanoalanine by an enzyme called β-cyanoalanine synthase (CAS) has been best studied in bacteria and plants that need to protect themselves from HCN during the synthesis of cyanogenic glucosides or ethylene. The enzymes responsible for CAS activity also have cysteine synthase activity (CYS) and are referred to as CYS or CAS depending on their substrate specificity. CYS catalyzes the conversion of *O*-acetylserine into cysteine, an essential final step in the cysteine biosynthesis pathway unique for plants and bacteria (*Bonner et al., 2005*). Animals synthesize cysteine by a different pathway and use related enzymes such as cystathionine-β-synthase (CBS) and cystathionine-γ-lyase (CGL), which form together with CAS and CYS a group of pyridoxal-5′-phosphate dependent enzymes (*Finkelstein et al., 1988*; *Figure 1*). Here we identify the enzyme responsible for detoxification of cyanide to β-cyanoalanine in a spider mite and show that this enzyme is horizontally acquired from bacteria and is also widely distributed in Lepidoptera.

## Results

### Transcriptional response of the spider mite *Tetranychus urticae* to a cyanogenic host plant

In order to gain a better insight into arthropod defenses against plant cyanogenesis, we used the two-spotted spider mite *T. urticae* as a generalist herbivore model in a host plant adaptation experiment.

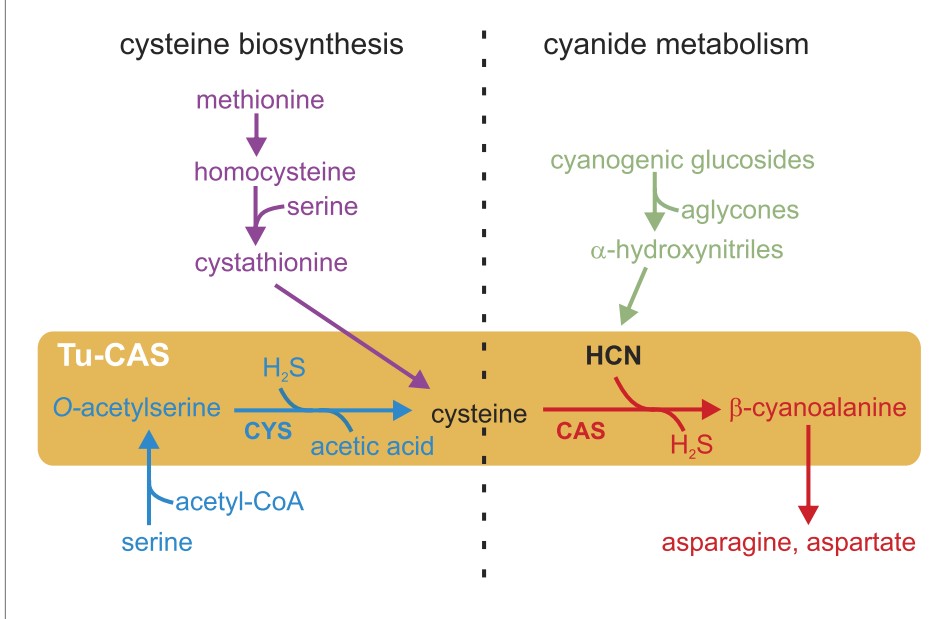

**Figure 1**. Schematic overview of the cysteine biosynthesis pathway in Metazoa (purple) and Plants/Bacteria (blue), the release of HCN during plant cyanogenesis (green) and the main HCN detoxification pathway in arthropods (red). The two reactions catalyzed by the gene product of the *Tu-CAS* gene are marked by an orange background and are indicated by CYS and CAS. CAS detoxifies cyanide by incorporation into cysteine forming β-cyanoalanine, which can be further metabolized by nitrilases. CYS catalyzes the second step of the cysteine synthesis pathway in bacteria and plants, after serine is converted to *O*-acetylserine.

This species is one of the most polyphagous arthropod pests known to date, and feeds on more than 1100 plant species from more than 140 plant families, including many cyanogenic plants (*Grbic et al., 2011*). We transferred a spider mite strain reared on acyanogenic bean plants (*Phaseolus vulgaris*) to a cultivar of *Phaseolus lunatus* containing high levels of well characterized cyanogenic glucosides such as linamarin and lotaustralin (*Ballhorn et al., 2006*; *Jones, 1998*; *Wybouw et al., 2012*) and allowed the strain to adapt to this host plant for more than 30 generations. Gene expression differences between mites feeding on *P. vulgaris* and on *P. lunatus* were then determined using genome-wide microarrays (*Dermauw et al., 2013*).

In contrast to the broad response previously detected after host plant changes from Fabaceae to Solanaceae and Brassicaceae (*Grbic et al., 2011*; *Dermauw et al., 2013*), only a limited set of 28 genes (absolute fold change ≥2, Benjamini-Hochberg corrected p-value <0.05) was found differentially expressed between the parental and adapted lines (*Supplementary file 1*). Within this small set of genes, 18 had an increased expression in the adapted line, while 10 exhibited a lower expression. The most differentially expressed genes after transfer to cyanogenic lima bean encode closely related, small cytoplasmic proteins of unknown function. They were also seen in the transcriptional response after transfer to tomato suggesting that these genes could be part of a broad general stress response of *T. urticae* (*Dermauw et al., 2013*). Among the genes with an increased expression were three cytochrome P450 genes, known to respond readily to host plant changes (*Grbic et al., 2011*; *Dermauw et al., 2013*). Moreover, we identified a gene (*tetur10g01570, Tu-CAS*) encoding a predicted cytosolic protein with high similarity to bacterial cysteine synthases (Conserved Domain Database (CDD): COG0031). The reported microarray data have been deposited at the Gene Expression Omnibus (*Wybouw et al., 2013*).

## Phylogeny of Tu-CAS and evidence for a bacterial origin by horizontal transfer

Within chelicerates, we only detected close homologues of *Tu-CAS* in two closely related tetranychid mites. By sequencing PCR amplicons (see below, this section) and a tBLASTn-search in a published transcriptome (*Liu et al., 2011*), *CAS* genes were identified in *Tetranychus evansi* and *Panonychus*

citri, respectively. A tBLASTn-search in published genomes of mesostigmatid mites and of ticks (Metastigmata) did not reveal homologues (**Supplementary file 2**). Broadening the search to arthropods by tBLASTn-searches in NCBI databases and additional arthropod genome portals (**Supplementary file 2**), we only detected close homologues of Tu-CAS in lepidopteran genomes (*Bombyx mori*, *Danaus plexippus*, *Heliconius melpomene*, *Manduca sexta* and *Plutella xylostella*) (**Figure 2**). On average, Tu-CAS showed 75% similarity with lepidopteran protein sequences, and genes encoding these proteins were intronless in both mites and insects. By searching additional NCBI EST-databases and published lepidopteran transcriptomes, a total of 20 (complete and partial) homologous sequences were identified in arthropods.

A phylogenetic reconstruction of these arthropod proteins with CYS, CAS and CBS enzymes of plants, fungi, oomycetes, bacteria and Metazoa indicated that these homologous arthropod sequences might be monophyletic. They were embedded with high node support within bacterial cysteine synthase sequences, indicative of a horizontal gene transfer (**Figure 2**). The most closely related sequences are from bacteria that belong to the α- and β-Proteobacteria, two of which are *Methylobacterium* species, free-living epiphytic bacteria known to establish endophytic colonies.

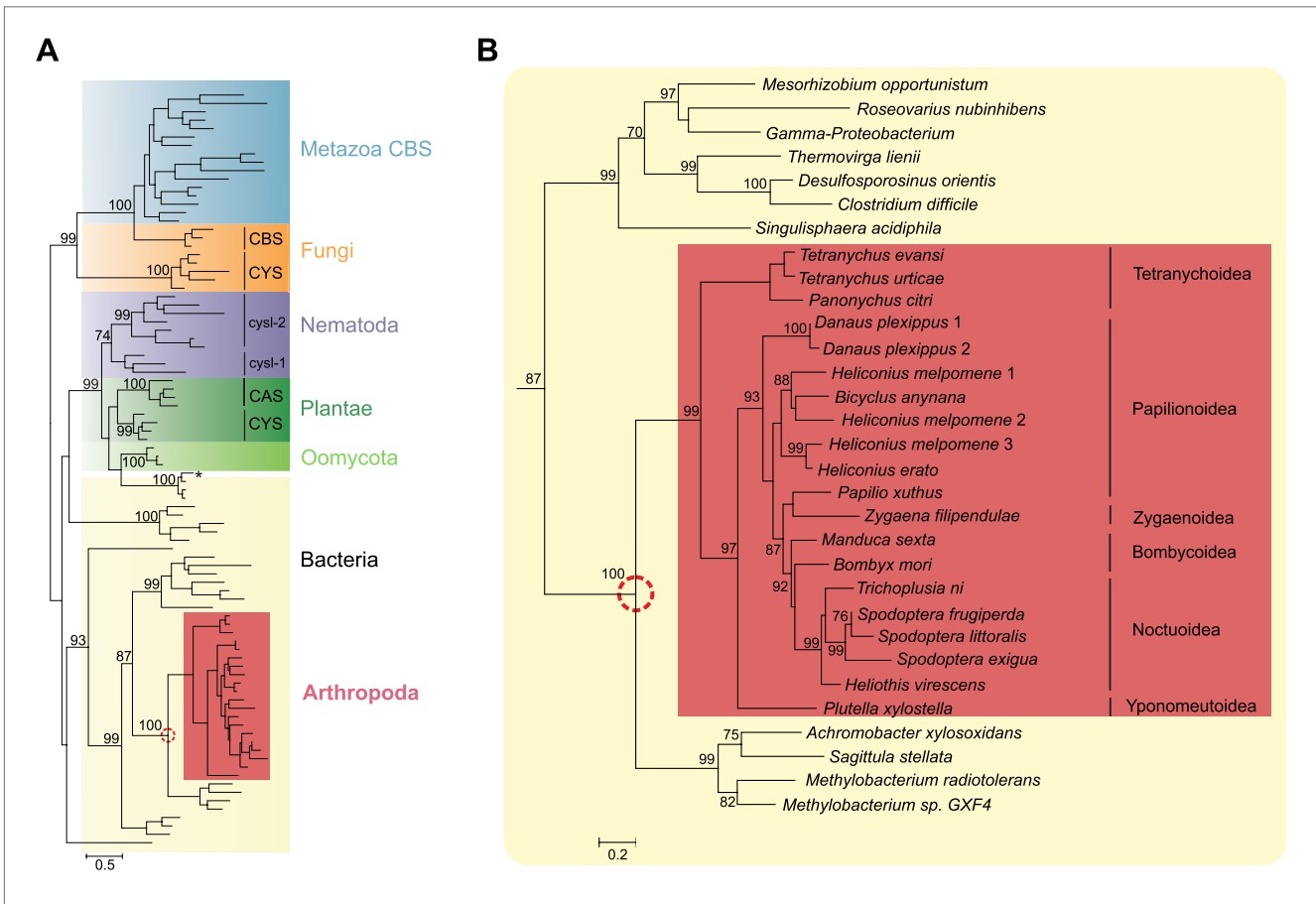

**Figure 2**. Panel A: Phylogenetic analysis of β-substituted alanine synthases, showing arthropod sequences nested within bacterial cysteine synthases. The fungal CYS, metazoan and fungal CBS as well as the plant, oomycete and nematode CYS and CAS sequences are marked with a different color. The two branches of nematode sequences, marked as cysl-1 and cysl-2, include the sequences coded by the two genes previously characterized in *C. elegans* (**Budde and Roth, 2011**). The CYS and CAS groups within Plantae represent plant protein sequences with CYS and CAS activity, respectively (**Yamaguchi et al., 2000**). The asterisk represents a CYS sequence of the mealybug *P. citri* acquired by horizontal gene transfer from its endosymbiont (**Husnik et al., 2013**). Panel **B**: Detailed view of the bacterial CYS sequences showing the embedded sequences of tetranychid mites and Lepidoptera. In both panels support values of only important nodes are shown. The scale bar represents 0.5 and 0.2 substitutions per site in panel **A** and panel **B**, respectively.

The following figure supplements are available for figure 2:

**Figure supplement 1**. MUSCLE alignment of cysteine synthases and β-cyanoalanine synthases discussed in this study.

These species are reported to be transferred from the plant to phytophagous insects and to survive inside arthropod hosts (*Kutschera, 2007*; *Rampelotti-Ferreira et al., 2010*), facilitating a potential horizontal gene transfer. Alternatively, Proteobacteria are known endosymbionts of arthropods and often reside in the reproductive organs for vertical transmission to following generations (*Wernegreen, 2002*). Because of this intimate relationship, successful horizontal gene transfer is more likely to occur (*Hotopp, 2011*). The mite and insect sequences formed a branch in bacterial cysteine synthase enzymes, some of which have documented dual CYS and CAS activities (*Omura et al., 2003*) (*Figure 2*). The arthropod protein sequences and their closest bacterial homologues shared a unique 9 amino acid insertion not present in cysteine synthases of other organisms, but the residues known to be crucial for substrate and cofactor binding in plants and bacteria showed conservation (*Bonner et al., 2005*; *Yi et al., 2012*). The lysine residue (Lys95, *Glycine max* CAS numbering, *Yi et al., 2012*) that forms a Schiff base linkage to the cofactor pyridoxal-5′-phosphate was also conserved in arthropods (*Figure 2—figure supplement 1*).

To exclude the possibility that the ***Tu-CAS*** sequence was derived from contaminating bacterial DNA, we examined its position in the *T. urticae* genome into more detail. *Tu-CAS* is located on a 3 Mb large scaffold (scaffold 10) and is flanked by typical eukaryotic genes, *tetur10g01550* and *tetur10g01580*. Both genes contain introns with splice sites confirmed by EST or RNA-seq data (*Grbic et al., 2011*, GenBank: LIBEST_025606, *Figure 3*). *Tetur10g01580* encodes a nudix hydrolase (CDD: cl00447) highly similar to other arthropod nudix hydrolases (BLASTp hits with E-value $<1e^{-45}$). To rule out that the linkage of *Tu-CAS* to these spliced genes is a genome assembly artefact, we took several independent genomic approaches. First, we remapped the *T. urticae* Sanger reads used for the genome assembly of the London strain of *T. urticae* (*Grbic et al., 2011*) and examined the Illumina read coverage of the *Tu-CAS* region in two re-sequenced strains (EtoxR and Montpellier) that are geographically distinct from the London strain (*Grbic et al., 2011*; *Van Leeuwen et al., 2012*). Neither Sanger nor Illumina-reads revealed any inconsistencies in the *Tu-CAS* region in any of these strains (*Figure 3*). Second, using a PCR approach, we amplified a 6 kb genomic region bracketing *Tu-CAS* with the surrounding spliced genes (*tetur10g01550* and *tetur10g01580*, *Figure 4*, *Figure 4—figure supplement 1*). Using a similar strategy we also amplified a 6 kb region in the closely related spider mite *T. evansi* (*Figure 4*, *Figure 4—figure supplement 1*) and a nucleotide dot plot between the amplified region of these two species showed clear synteny and the absence of discontinuity around the *CAS* gene (*Figure 4*). This would not be expected by bacterial contamination of either genome. Last, gene compositions of *T. urticae* and bacteria were analyzed by determining both the GC-content at the synonymous third codon position (GC3) and the overall GC-content (GC) of the genes to look if amelioration occurred. Amelioration is the process by which the DNA composition of the newly acquired gene becomes homogenized to match the composition of the recipient genome (all genes in the recipient genome are subject to the same mutational processes) (*Lawrence and Ochman, 1997*). Indeed, the GC/GC3 content of *Tu-CAS* was most similar to the GC/GC3 content of genes from the *T. urticae* genome, and quite distinct from the GC/GC3 content of genes (including the *CYS/CAS* genes) from the three anno-tated bacterial genomes in the closest sister clade to the apparent monophyletic arthropod clade (*Figure 5*). Taken together, these data provide strong evidence that *Tu-CAS* is a sequence integrated in the genome of *T. urticae* and does not represent bacterial contamination. Sequence data is available at Genbank (accession numbers: KF981736 and KF981737).

## Biochemical characterization of Tu-CAS

In order to functionally test whether arthropod enzymes are active and still able to catalyze both reactions after horizontal gene transfer, we recombinantly expressed *Tu-CAS* in *Escherichia coli* and obtained mg quantities after affinity purification. Subsequent biochemical assays confirmed that Tu-CAS catalyzes both reactions (*Figure 6*). In order to determine which of the two reactions (cysteine synthesis, CYS and β-cyanoalanine synthesis, CAS) was favored by the Tu-CAS enzyme, we calculated the ratio of the specificity constants for each reaction. The specificity constant $k_{cat}/K_m$ defines, at any concentration, the specificity of an enzyme for a particular substrate. The CAS/CYS ratio of the speci-ficity constants was 33.7, showing that CAS activity was strongly favored over CYS activity (*Table 1*). (The ratio was calculated from the respective values of $V_{max}/K_m$, as $k_{cat} = V_{max}/[E]$ and as the reactions were measured with the same enzyme preparation.) This preferred CAS activity is typical of known plant CAS enzymes and clearly different from CYS enzymes (*Table 1*; *Yamaguchi et al., 2000*; *Wada et al., 2004*; *Bogicevic et al., 2012*; *Yi et al., 2012*). Enzymatic activity was dependent on

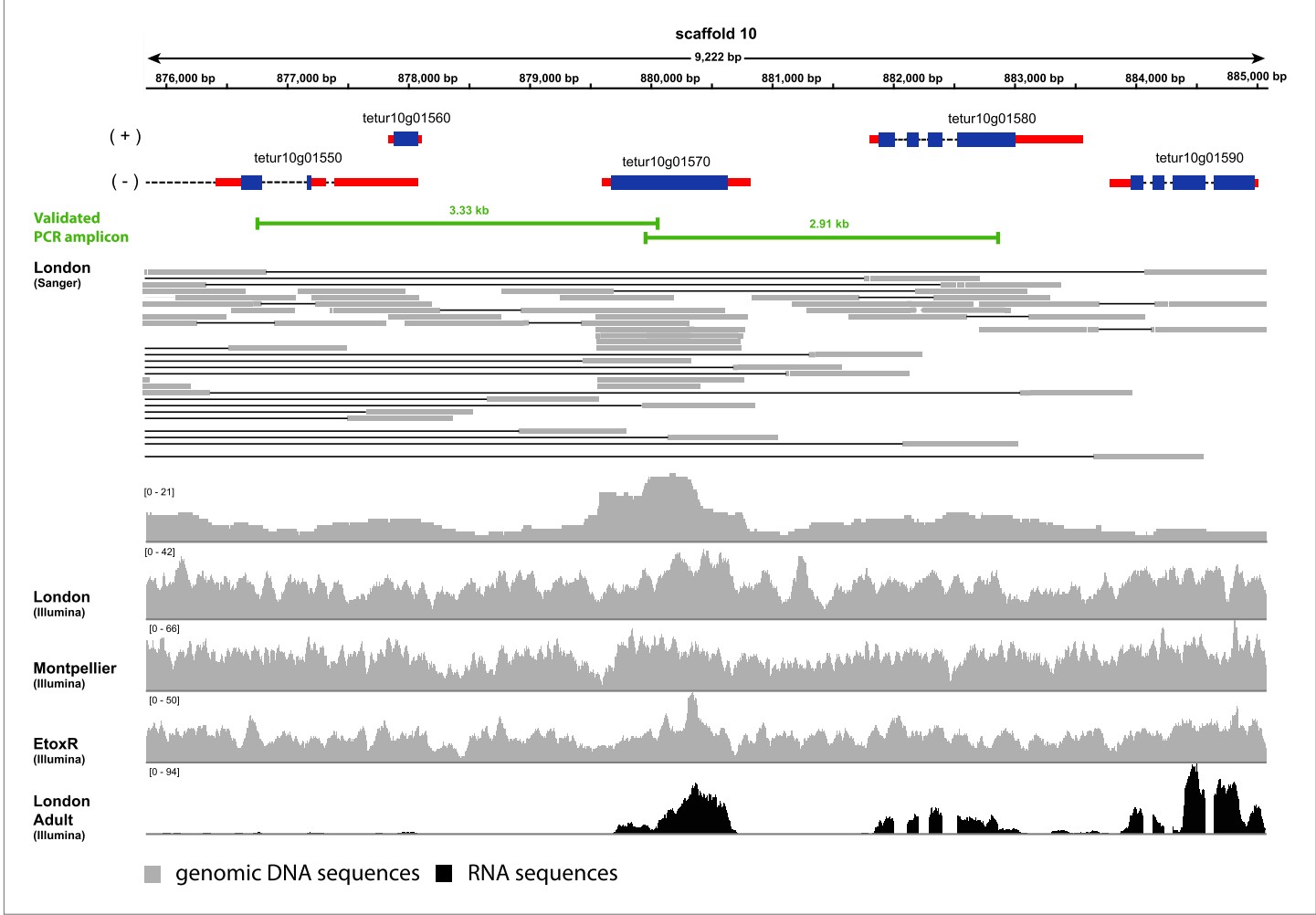

**Figure 3**. Coverage plot of *Tu-CAS* (*tetur10g01570*) and its surrounding region in the genome of *T. urticae*. Gene models of *Tu-CAS* and its neighboring genes are depicted as follows: blue and red rectangles represent coding sequences and untranslated regions, respectively, while introns are shown as dashed lines. (+) and (−) represent the forward and reverse strand, respectively. Underneath the gene models, indicated in green, are the length and position of amplicons obtained by PCR (*Figure 4*). Next, an alignment of paired-end Sanger reads (and corresponding coverage plot) with the *T. urticae* genome of the London strain is displayed. Paired-end Sanger reads for which both reads are mapped in or extend nearby the indicated region are denoted by thin lines to show pair connections (shown are all paired-end Sanger reads that were produced from 2.5, 8.5, and 35.5 kb libraries used for assembly of the *T. urticae* genome [*Grbic et al., 2011*]). The Sanger reads coverage plot is followed by coverage plots of Illumina-reads from genomic DNA sequencing of the Montpellier and EtoxR strain of *T. urticae* (*Grbic et al., 2011*; *Van Leeuwen et al., 2012*). The coverage plot at the bottom shows Illumina RNA-seq read coverage produced from adult *T. urticae* polyA selected RNA (*Grbic et al., 2011*). Numbers between brackets represent the sequence depth.

pyridoxal-5′-phosphate as a cofactor, and the substrate-dependent formation of β-cyanoalanine was further confirmed by thin layer chromatography (TLC) and LC-MS (*Figure 7*).

The very high sequence similarity between Tu-CAS and the lepidopteran proteins strongly suggests that the lepidopteran proteins can catalyze the same two reactions (cyanide detoxification and cysteine synthesis) and that these proteins are responsible for the known wide occurrence of CAS activity in lepidopteran species (*Witthohn and Naumann, 1987*; *Meyers and Ahmad, 1991*; *Stauber et al., 2012*).

## Discussion

In arthropods, the ability to detoxify HCN plays a crucial ecological role and is thought to have allowed the exploitation of cyanogenic plants by circumventing the toxic effects of HCN. We have shown here that the two-spotted spider mite increases transcript levels of a horizontally transferred β-cyanoalanine synthase upon adaptation on cyanogenic bean. Phylogenetic evidence alone does not constitute

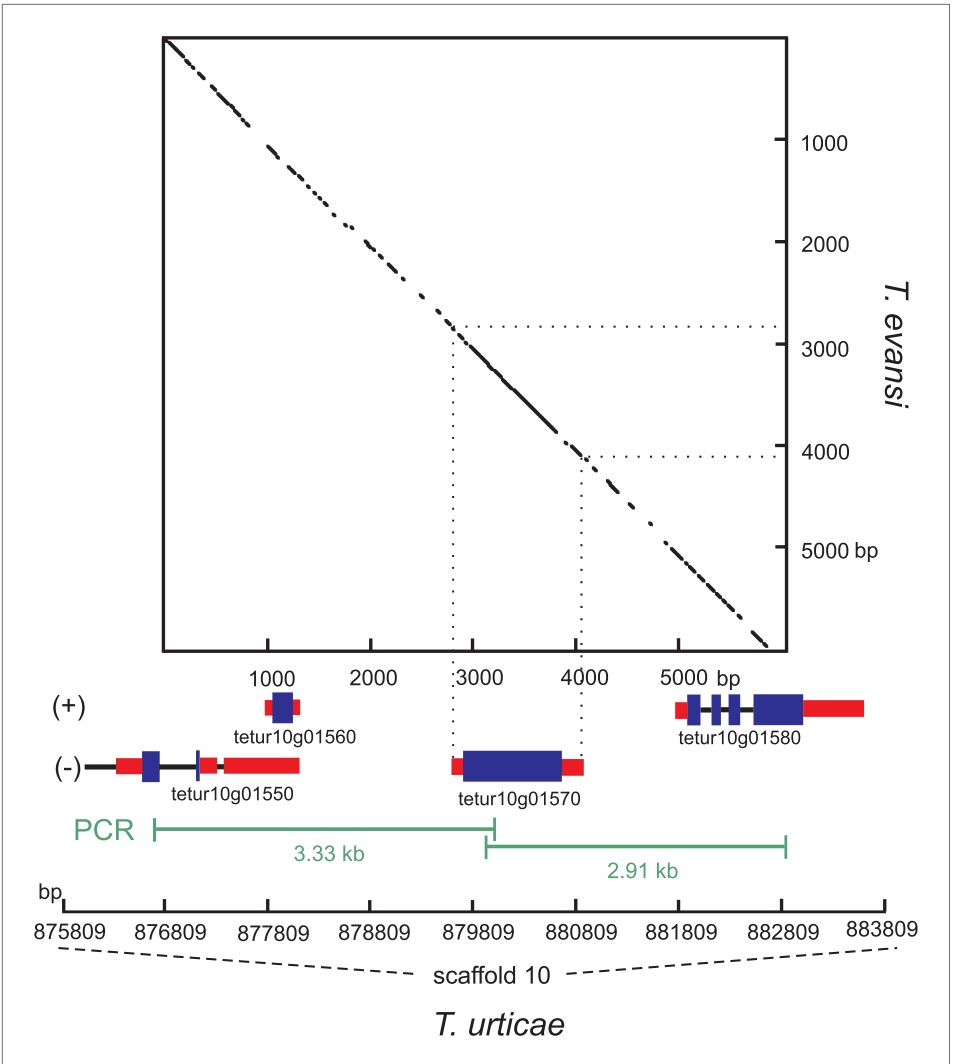

**Figure 4**. Nucleotide dot plot of the PCR amplified genomic region bracketing *T. urticae* and *T. evansi CAS* with adjacent intron-containing eukaryotic genes. The dot-plot was constructed with 95% identity in a 21 bp window, with the *T. evansi* and the *T. urticae* amplified region on the y- and x-axis, respectively. From the *T. urticae* region, the gene models and their genomic positions on the 10ᵗʰ scaffold are specified below the x-axis. The (+) and (−) signs represent the forward and reverse strand, respectively. Blue and black bars indicate exons and introns respectively, while the untranslated regions are depicted as red bars.

The following figure supplements are available for figure 4:

**Figure supplement 1**. Agarose gel of PCR products, bracketing *CAS* with adjacent eukaryotic genes in *T. urticae* and *T. evansi*.

strong evidence for horizontal gene transfer (HGT), because in the absence of introns in the sequence, a contamination of the mite genome sequence with a sequence from a bacterial symbiont or commensal cannot be excluded. However, the results of mite genomic analysis, codon amelioration, synteny and genomic PCR, combined with the phylogenetic evidence, unambiguously prove that *T. urticae CAS* has a bacterial origin and was laterally transferred and incorporated into the mite genome prior to the divergence in the Tetranychidae. A homologous lateral gene transfer has also occurred in Lepidoptera (*Figure 2*) raising the question of the time and number of HGT events needed to explain the phylogenetic pattern of distribution of the *CYS/CAS* genes. Several hypotheses can be discussed. (a) A single HGT to a common ancestor of mites and insects, followed by selective losses resulting in the present phylogenetic distribution. The broad sampling of arthropod species that

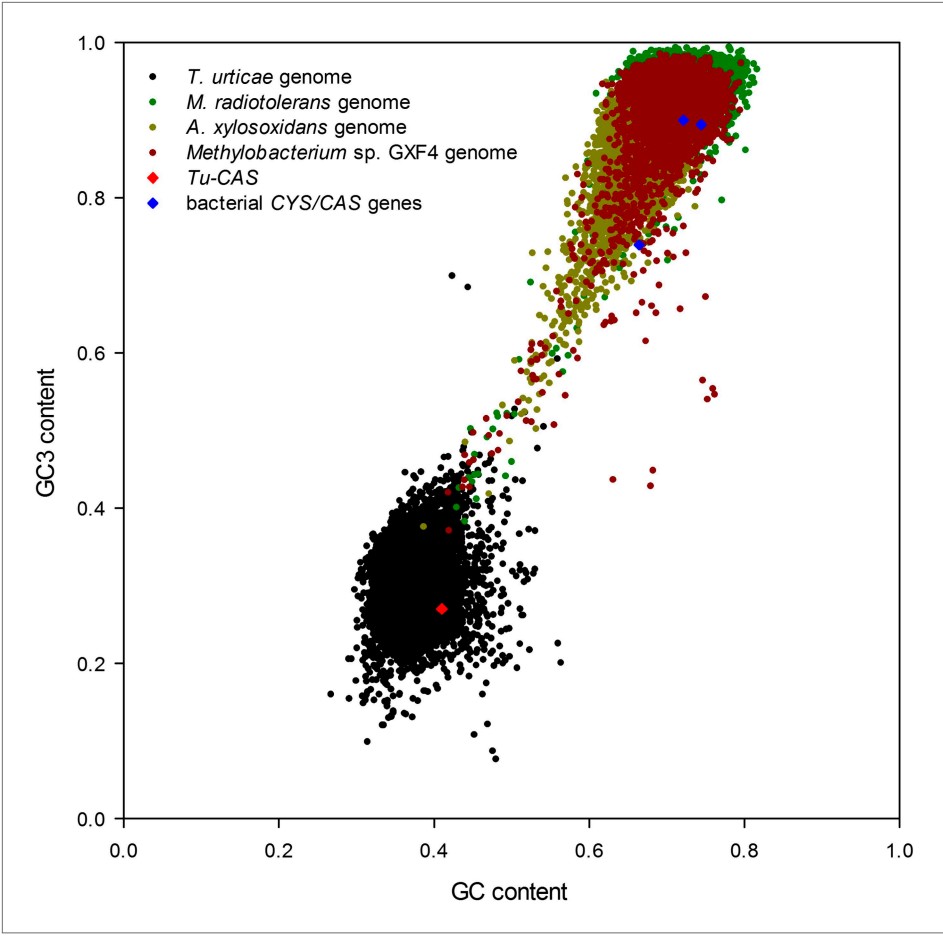

**Figure 5**. Graph showing the GC/GC3 gene contents of *T. urticae* and putative bacterial donor species. The GC/GC3 contents of the *T. urticae* genome and the three annotated bacterial genomes of bacteria residing in the sister clade closest to the apparent monophyletic arthropod clade (*Figure 2*) are shown. The GC/GC3 content of the specific *CYS/CAS* gene of each organism (*T. urticae* in red, bacteria in blue) is highlighted.

revealed the absence of any *CYS/CAS*-like sequence (even as a distant recognizable trace) argues against the origin of the gene in the common ancestor of tetranychid mites and Lepidoptera, followed by multiple independent losses. These losses would have to be numerous: seven in the hexapods (Trichoptera, Diptera, Hymenoptera, Coleoptera/Strepsiptera, Hemiptera/Heteroptera/Thysanoptera/ Phthiraptera, Orthoptera, Odonata/Ephemeroptera) and six further in other arthropods (Copepoda/ Branchiopoda, Myriapoda, Aranea, Scorpiones, Metastigmata and Mesostigmata). This figure of 13 losses is an absolute minimum that implies that the loss occurred each time at the origin of the lineage, that is before any speciation beyond the point of coalescence. A loss later in the history of each lineage would rapidly increase the total number of losses. We strongly believe that this hypothesis is not parsimonious, and therefore that there was no *CYS/CAS* gene in the common ancestor of mites and Lepidoptera. (b) Alternatively, two HGT events might have occurred (one to a mite ancestor, one to a lepidopteran ancestor) or (c) a single HGT from a donor bacterial species, followed by a transfer between a mite and a lepidopteran. These two hypotheses are impossible to distinguish based on the topology of the tree (*Figure 2*). The sampling of bacterial species phylogenetically close to the presumed donor species is too shallow at present. The apparent monophyly of the arthropod *CAS* gene may be due to the fact that both the mite and the insect gene came from very closely related bacteria, but these bacteria are not represented in the tree. A future survey of Proteobacteria likely to be associated with arthropods (directly or through a plant host) may resolve this question by finding several potential bacterial donor species that would invalidate the apparent monophyly (i.e., split the tree at the point of arthropod coalescence). The transfer from a lepidopteran to mite is improbable,

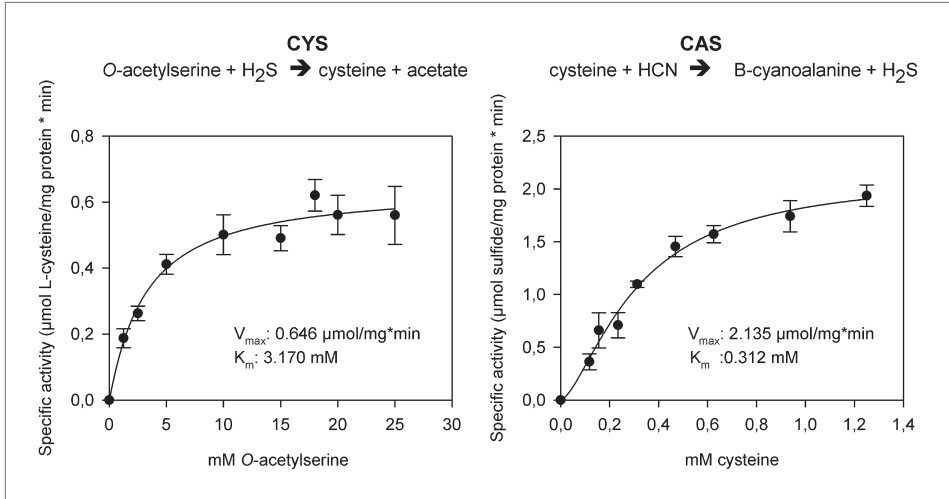

**Figure 6**. The two reactions catalyzed by recombinant Tu-CAS (cysteine synthase, CYS and β-cyanoalanine synthase, CAS), showing the kinetic plots and calculated $V_{max}$ and $K_m$ values.

because the mite sequence would branch with the closest relative to the lepidopteran donor, rather than being basal. Conversely, we cannot exclude the transfer from a mite to a lepidopteran, as sampling within the Prostigmata is extremely limited. We therefore favor hypothesis (b), with a very old transfer to an ancestral lepidopteran, and a second transfer to an ancestral mite. Current sampling of mite species is insufficient to give a good estimate of time, but we note the absence of a CAS enzyme in Mesostigmata (typically parasitic and predatory mites) and Metastigmata (ticks), and transfer is likely to have occurred after the split of Prostigmata in the Lower Devonian about 400 MYA (***Dabert et al., 2010***). The hypothesis of multiple transfers is also compatible with the presence of a *CYS*-like sequence in the mealybug *Planococcus citri*, apparently derived from its endosymbiont (***Husnik et al., 2013***). In that case, the phylogenetic tree clearly distinguishes the HGT event from that under study here.

**Table 1.** Specificity constants for the two activities of CYS-like enzymes (cysteine synthase, CYS and β-cyanoalanine synthase, CAS)

| | CAS reaction | | | CYS reaction | | | CAS/CYS |
|---|---|---|---|---|---|---|---|
| | Activity (s⁻¹) | $K_m$ (mM) | Specificity constant (mM⁻¹. s⁻¹) | Activity (s⁻¹) | $K_m$ (mM) | Specificity constant (mM⁻¹. s⁻¹) | Ratio of specificity constants |
| *Tetranychus urticae* CAS | 2.135* | 0.312 | 6.84† | 0.646* | 3.17 | 0.203† | **33.7** |
| *Arabidopsis thaliana* CAS | 2.66 | 0.14 | 19 | 2.0 | 8.03 | 0.250 | **76** |
| *Glycine max* CAS | 38.9 | 0.81 | 48 | 1.82 | 8.87 | 0.205 | **234** |
| *Glycine max* CYS | 0.21 | 0.30 | 0.7 | 57.5 | 3.6 | 15.97 | **0.044** |
| *Corynebacterium glutamicum* CYS | n.d | n.d | – | 435* | 7 | 62† | – |
| *Lactobacillus casei* CYS | n.d | n.d | – | 89* | 0.6 | 148† | – |

n.d.: not determined.
*data as $V_{max}$ in μmol.min⁻¹mg⁻¹.
†data as $V_{max}/K_m$ in μmol.min⁻¹mg⁻¹mM⁻¹. CAS activity was measured with cysteine as substrate while CYS activity was measured with *O*-acetylserine as substrate. Data for the plants *A. thaliana* and *G. max* were obtained in ***Yamaguchi et al. (2000)*** and ***Yi et al. (2012)***, respectively, while data for the bacteria *C. glutamicum* and *L. casei* were retrieved from ***Wada et al. (2004)*** and ***Bogicevic et al. (2012)***, respectively.

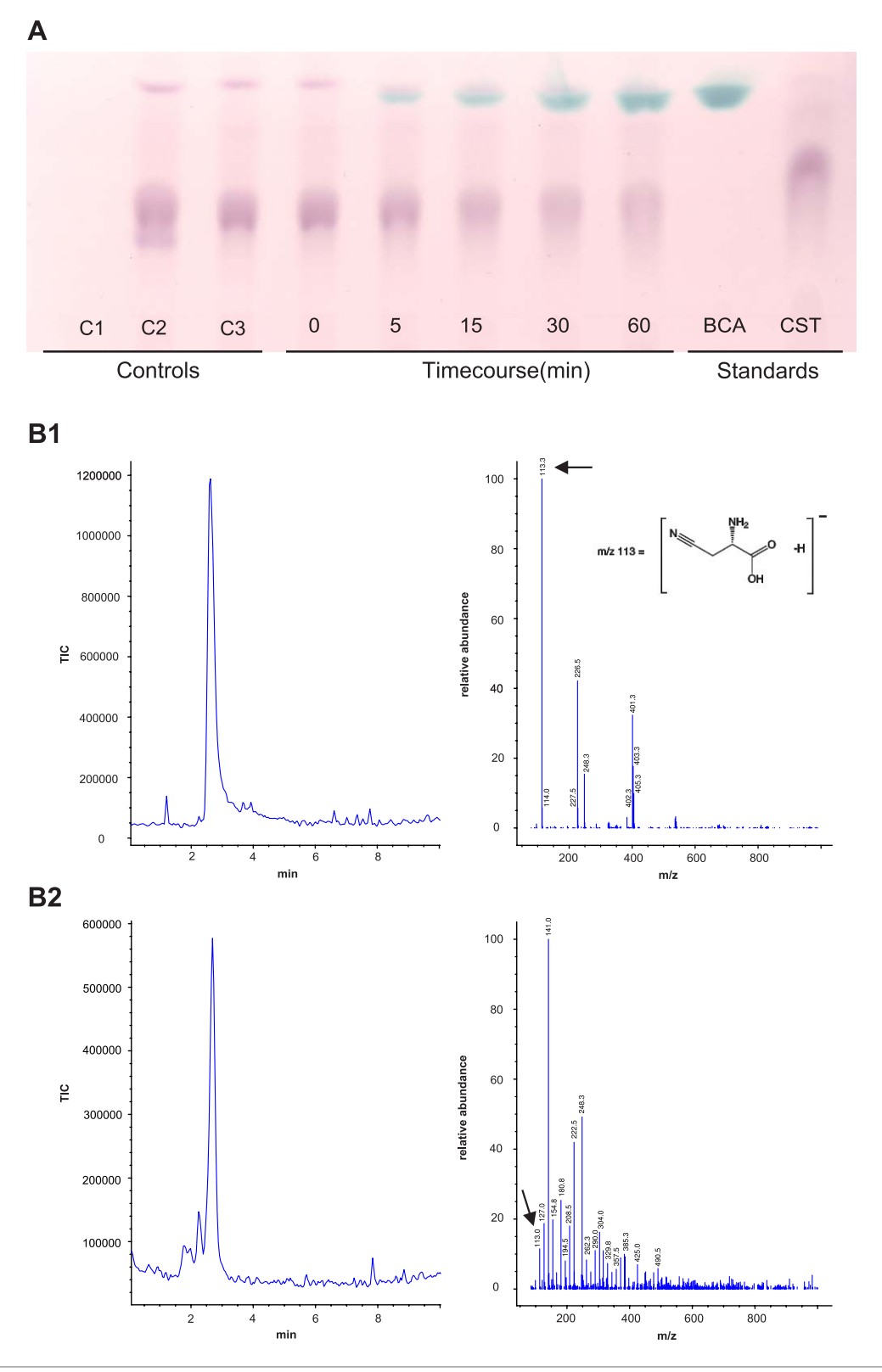

**Figure 7**. Panel A: Formation and accumulation of β-cyanoalanine by recombinant Tu-CAS as visualized by TLC analysis. Controls; C1: no cysteine control, C2: no cyanide control, C3: no enzyme control. Time course 0–60 min after adding 3.75 μg of recombinant Tu-CAS. Standards; BCA: 5 μg β-cyanoalanine, CST: 5 μg cysteine.
*Figure 7. Continued on next page*

*Figure 7. Continued*

Panel **B**: LC-MS identification of β-cyanoalanine as a reaction product of Tu-CAS. The enzymatically produced β-cyanoalanine was scraped from silica plates after TLC separation of reaction mixtures, and was analyzed by LC-MS. β-cyanoalanine was identified in the reaction mixture on the basis of a similar elution time on LC and the characteristic ion of m/z = 113 which is [M-H]⁻ as compared to the standard. (The base peak in panel B2 at m/z = 141 is a contaminant from the silica gel [$2SiO +2H_2O + OH^-$].) **B1**: total ion current (TIC) chromatogram and mass spectrum of the β-cyanoalanine standard, **B2**: TIC chromatogram and mass spectrum of Tu-CAS reaction mixture after separation on TLC.

In Lepidoptera, the *CAS* gene was subsequently duplicated in *H. melpomene* and *D. plexippus* that have 3 and 2 copies of the gene, respectively (*Sun et al., 2013*). The importance of the laterally acquired gene for HCN detoxification in those species was not previously apparent. These species of Papilionoidea, like burnet moths, not only thrive on cyanogenic plants, but have themselves evolved the ability to synthesize cyanogenic compounds de novo (*Jensen et al., 2011*) that now have crucial functions in their life history. Cyanogenesis serves in predator defense by releasing HCN, but also stores reduced nitrogen that can be mobilized for chitin synthesis, and plays a role in mate choice by determining the attractiveness of nuptial gifts from male to female partners (*Zagrobelny et al., 2007*). Our results support the early idea that CAS activity is needed for the exploitation of cyanogenic glucosides in insects (*Meyers and Ahmad, 1991*; *Witthohn and Naumann, 1987*; *Zagrobelny et al., 2008*) whether they are sequestered from the plant, or synthesized de novo. Higher β-cyanoalanine synthase activity in *Spodoptera eridania* than in *Trichoplusia ni* is related to higher cyanide tolerance (*Meyers and Ahmad, 1991*), and this enzyme activity is widespread in Lepidoptera (*Witthohn and Naumann, 1987*).

Moreover, it was recently shown that specialist pierid butterflies that feed on Brassicales, release equimolar concentrations of HCN upon metabolism of benzylglucosinolates, turning the 'mustard oil bomb' into a 'cyanide bomb' (*Stauber et al., 2012*). When *Pieris rapae* feeds on a cyanogenic (dhurrin-containing) plant that this species does not normally consume, an increased production of β-cyanoalanine and thiocyanate is observed, thus implicating both a CAS and a rhodanese activity (*Stauber et al., 2012*). It was therefore proposed that the ability of *P. rapae* to metabolize HCN allowed the primary host transfer from Fabales to Brassicales (*Stauber et al., 2012*). The gene for either CAS or rhodanese has not been identified in arthropods before, and their respective role in detoxification of HCN is not formally demonstrated. There is no close homologue of the known rhodanese (thiosulfate sulfurtransferase) gene in *T. urticae* or in Lepidoptera. However, we identified *Tu-CAS* and functionally demonstrated that the enzyme it encodes converts HCN to β-cyanoalanine in vitro. Such evidence is difficult to obtain in vivo with mites, as it would be difficult to exclude the possibility that a plant or a bacterial enzyme rather than the mite enzyme catalyzes the reaction in vivo. We argue that the presence of the same gene in lepidopteran species that display this activity in vivo (*Meyers and Ahmad, 1991*; *Stauber et al., 2012*) is strong evidence for the function of the laterally transferred *CAS* gene. It will be of great interest to confirm that the homologous *CAS* genes of Lepidoptera that we identified indeed encode a β-cyanoalanine synthase, and to provide evidence of its protective role against HCN poisoning.

Next to the detoxification function, the CYS activity acquired after horizontal gene transfer may also have enhanced the sulfur amino acid economy of mites and lepidopterans (*Figure 1*). To date, nematodes were the only animal species thought to synthesize cysteine independently from methionine by CYS activity (*Budde and Roth, 2011*). However, nematode CYS sequences grouped with plants and oomycetes, clearly outside the arthropod-bacterial clade, suggestive of a different origin of CYS between metazoan subgroups (*Figure 2*). Duplications of *cys* genes were observed in nematodes, and indeed genetic evidence suggests that in *Caenorhabditis elegans* cyanide resistance is conferred by the *cysl-2* gene, probably encoding an enzyme with CAS activity while *cysl-1* is a classical cysteine synthase gene (*Budde and Roth, 2011*). The acquisition of an alternative cysteine biosynthesis route fits into previously documented horizontal gene transfers in *T. urticae* that include a cobalamin-independent methionine synthase gene, genes for carotenoid biosynthesis, as well as laterally acquired genes that likewise respond to host plant change such as intradiol ring-cleavage dioxygenases and a cyanase gene (*Grbic et al., 2011*; *Wybouw et al., 2012*; *Dermauw et al., 2013*). The latter encodes an enzyme that decomposes cyanate (CNO⁻), a bacterial or photochemical decomposition product of cyanide, to carbon dioxide and ammonia (*Wybouw et al., 2012*). This enzyme may serve as a second line of spider mite defense against cyanogenic plants, or alternatively may have a regulatory function

in the amino acid and pyrimidine metabolism as previously suggested (*Wybouw et al., 2012*). For a polyphagous herbivore, horizontal gene transfer might play an important role in gaining independence from the varying plant nutrients and defense compounds. It remains unclear which reaction (CAS/CYS) of Tu-CAS provides the strongest adaptive advantage, but the CYS activity might be one of the reasons why these horizontally transferred genes have been retained in organisms that are at present not living on cyanogenic plants.

In conclusion, a horizontal gene transfer from a bacterial ancestor underlies the exploitation of cyanogenic host plants in some arthropod lineages and made the subsequent evolution of a convergent pathway for synthesis of cyanogenic glucosides possible, as shown in burnet moths (*Jensen et al., 2011*).

## Material and methods

### Mite strains and host plants

The London strain of *T. urticae* (*Grbic et al., 2011*) was maintained on acyanogenic *P. vulgaris* L. cv 'Prelude'. The strain London-CYANO originated from this population and was transferred to cyanogenic *P. lunatus* as previously described (*Wybouw et al., 2012*). Before the start of experiments the cyanogenic potential of both plant species was determined, confirming the acyanogenic nature of *P. vulgaris* and revealing high levels of cyanogenic precursors produced in the *P. lunatus* cultivar (*Wybouw et al., 2012*). For this study, young adult female mites were collected for gene-expression analysis 35 generations after the initial host shift. The *T. evansi* strain was maintained in the laboratory on *Solanum lycopersicum* L. cv 'Moneymaker' as previously described (*Wybouw et al., 2012*). All strains were maintained in climatically controlled rooms at 26°C, 60% RH and 16:8 hr light:dark photoperiod.

### Transcriptional response of *T. urticae* to a cyanogenic host plant

Total RNA samples were isolated with the RNeasy minikit (Qiagen, Belgium) and were subsequently treated with DNase (Turbo DNA-*free* kit, Ambion, Belgium). RNA was extracted from 100–120 young adult female mites in four replicates. Cy5- or Cy3-labeled cRNA was produced using the Low Input Quick Amp Labeling Kit (Agilent Technologies, Belgium) as previously described (*Dermauw et al., 2013*). Microarray hybridization and scanning procedures were performed as previously described (*Dermauw et al., 2013*), using the GE2_107_Sep09 protocol. The data was transferred to GeneSpring GX 11.0 software (Agilent Technologies) for statistical analyis. Probes were flag filtered (only probes that had flag-value 'present' in 50% of all replicates were retained) and linked to *T. urticae* genes using the 'Create New Gene-Level Experiment' option. Differentially expressed genes were identified by a Student's *t* test with the cutoff for Fold Change (FC) and corrected p-value (Benjamini-Hochberg correction) set at 2 and 0.05, respectively. The array design is accessible under the GEO-platform format GPL16890 (*Wybouw et al., 2013*).

### Phylogenetic reconstruction

A full-length *Tu-CAS* (*tetur10g01570*) homologue was retrieved in *T. evansi* by sequencing PCR products bracketing the *CAS* gene with neighboring genes (*Supplementary file 4*). Additional Tu-CAS homologues were identified by conducting BLASTp and/or tBLASTn searches in NCBI, UniProt, *P. citri* transcriptome (*Liu et al., 2011*) and diverse arthropod genome portals (*Supplementary file 2*) using Tu-CAS as query. As several best BLAST-hits (E-value $\leq 1e^{-90}$) included members of the order Lepidoptera, transcriptome databases from Lepidoptera not included in the NCBI database were also mined for Tu-CAS homologues (*Supplementary file 3*). This approach yielded 35 arthropod and bacterial protein sequences, which according to the Conserved Domain Database all contained a motif typical for cysteine synthases (COG0031) (*Marchler-Bauer et al., 2011*). This dataset was further complemented with cysteine synthase M (cysM) protein sequences from bacteria and cysteine synthase protein sequences from fungi, Chromalveolata, plants, nematodes, and *Planococcus citri* and its three best BLASTp hits, harboring a cysteine synthase CDD motif (COG0031 or PLN2565). Finally, a diverse set of cystathionine-β-synthase protein sequences, related to cysteine synthases and also belonging to the group of pyrodixal-5′-phosphate dependent β-substituted alanine synthases were added as an outgoup.

The final dataset contained 90 protein sequences. Accession numbers of protein sequences, their trivial name, CDD classification (*Marchler-Bauer et al., 2011*) and cellular localization (*Horton et al., 2007*) are listed in *Supplementary file 3*. Protein sequences were aligned with MUSCLE (*Edgar, 2004*) using default settings. Model selection was done with ProtTest 2.4 and according to the Akaike information criterion the model LG+I+G was optimal for phylogenetic analysis (*Abascal et al., 2005*).

A maximum-likelihood analysis was performed using Treefinder (*Jobb et al., 2004*) with edge-support calculated by 1000 pseudoreplicates (LR-ELW). Resulting trees were midpoint rooted prior to further analysis (*Hess and De Moraes Russo, 2007*). Phylogenetic trees were visualized and edited using MEGA5 (*Tamura et al., 2011*) and CorelDraw X6 (Corel inc., UK), respectively.

## Incorporation of *Tu-CAS* in the spider mite genome

Paired-end *T. urticae* Sanger reads (available in the Trace Archive at the NCBI website, http://www.ncbi.nlm.nih.gov/Traces/home/) were remapped to the *T. urticae* genome (*Grbic et al., 2011*) using Bowtie 2.1.0 (*Langmead et al., 2009*) and the preset parameter option '–very-sensitive'. Resulting SAM files were converted into BAM files using SAMtools (*Li et al., 2009*). Illumina-reads from genomic DNA sequencing of the London, Montpellier and EtoxR strains of *T. urticae* and Illumina RNA-seq reads from adult *T. urticae* polyA selected RNA were mapped as previously described (*Grbic et al., 2011*; *Van Leeuwen et al., 2012*). Read alignments and coverage were visualized with IGV 2.3 (*Thorvaldsdottir et al., 2013*) using the most recent *T. urticae* genome annotation ('Tetur_gff3_20130708', accessible at http://bioinformatics.psb.ugent.be/orcae/-overview/Tetur); for display, Sanger reads were arranged in Adobe Illustrator CS5 while maintaining alignment coordinates.

Genomic DNA was collected from *T. evansi* and *T. urticae* by phenol-chloroform extraction (*Van Leeuwen et al., 2008*). Primer pairs were designed to amplify a genomic region of *Tu-CAS* and adjacent genes on either the 5′ or the 3′ end on the 10th scaffold of the *T. urticae* genome (*Supplementary file 4*). The Expand Long Range PCR kit (Roche, Belgium) was used to conduct PCR, and fragments were sequenced with primers listed in *Supplementary file 4*. Some primer pairs designed on the *T. urticae* genome sequence also successfully amplified genomic fragments of *T. evansi* (*Supplementary file 4*). The resulting fragments were sequenced by primer walking (*Supplementary file 4*). A nucleotide dot-blot between the two spider mite species was constructed using the MEGALIGN program of DNASTAR software, allowing 5% mismatch in a 21 bp window. We analyzed overall GC contents and at the third codon position (GC3) of whole coding nucleotide sequences using UGENE (*Okonechnikov et al., 2012*) of all coding sequences of *T. urticae* and *Achromobacter xylosoxidans*, *Methylobacterium radiotolerans* and *Methylobacterium* sp. GXF4. These three bacterial genomes were selected based on our phylogenetic analysis as the closest fully annotated bacterial genomes to the arthropod clade (*Figure 2*).

## Recombinant expression of Tu-CAS and enzyme activity assays

Recombinant Tu-CAS was produced by the GenScript Corporation (Piscataway, NJ, USA). After codon optimization of the *Tu-CAS* coding sequence (*Supplementary file 5*), an E3 expression vector was used to transform *E. coli* cells. The transformed cells were cultured in 3 l LB. Using a $Ni^{2+}$-column, the N-His-tagged Tu-CAS protein was purified from the supernatant. After sample sterilization via a 0.22 µm filter, the recombinant protein was stored in a buffer containing: 50 mM Tris, 150 mM NaCl, 2 mM DTT, 10% glycerol at a pH of 8.0 and finally kept at −80°C. The concentration and purity of the recombinant protein sample was determined respectively by a Bradford protein assay (*Bradford, 1976*) and a densitometric analysis of a Coomassie Blue-stained SDS-PAGE gel.

Chemicals for the activity assays were purchased from Sigma–Aldrich (Belgium), except β-cyanoalanine, which was acquired from VWR, Cayman Chemical. All reactions were carried out in gastight 7 ml vials with a screw cap having a PTFE/rubber septum (Supelco–Sigma–Aldrich, Belgium). Reagents were added to the reaction volumes using gastight syringes (Hamilton, series 1700, Gastight, 1750RN, VWR Belgium). Prior to measuring enzyme activity, recombinant Tu-CAS was incubated at 30°C for 10 min in the appropriate reaction buffer containing 500 µM pyridoxal-5′-phosphate. For measuring CAS activity, reactions were executed in a 0.2 M Tris buffer at pH 8.5. The CAS activity assay was a modification of the method of Hendrickson (*Hendrickson and Conn, 1969*). The standard reaction was started with 0.5 ml of 0.01 M cysteine, 0.5 ml of 0.01 M sodium cyanide and 1.5 µg recombinant Tu-CAS. All CAS assays were performed at 37°C on a mechanical shaker. CAS activity was quantified by spectrophotometrically measuring the $H_2S$ formed at 650 nm (PowerWavex340, BioTek Instruments Inc., Winooski, VT, USA) by the method of Siegel (*Siegel, 1965*). The CYS activity assay was based on *Lunn et al., 1990* using 0.5 µg recombinant Tu-CAS per reaction. Standard substrate concentrations were 10 mM and 2 mM for *O*-acetylserine and sodium sulfide, respectively. The reaction product cysteine was quantified by measuring the absorbance at 560 nm by the method of Gaitonde (*Gaitonde, 1967*). For both activity assays, the spontaneous formation of the measured reaction product and its potential presence in protein preparations was corrected by respectively non-enzyme controls and

zero time point controls. Each experimental assay condition was analyzed using three independent and three technical replicates. Kinetic data was fitted to the Hill equation, from which the $K_m$ and $V_{max}$ values for *O*-acetylserine and cysteine were calculated for respectively the CYS and CAS reaction.

## Identification of β-cyanoalanine by TLC and LC-MS

Recombinant Tu-CAS was incubated at 30°C for 10 min in a 1 mM phosphate buffer containing 500 μM pyridoxal-5'-phosphate. The standard reaction was executed in a 320 μl reaction volume at 37°C on a mechanical shaker in 1 mM phosphate buffer pH 8.5, with 80 μl of 7.5 mM cysteine and 80 μl of 15 mM sodium cyanide. Each reaction was started by adding 3.75 μg of recombinant protein and was terminated by snap freezing at different time points. No-substrate and no-enzyme controls were included in the analysis. Reaction mixtures were defrosted at 4°C and a 20 μl aliquot was spotted on a thin-layer chromatograph (HPTLC Silica gel 60 $F_{254}$, Merck, Darmstadt Germany) and run with a mobile phase of (ethanol/28% ammonium hydroxide/water) with a (18/1/4) ratio. Five μg of β-cyanoalanine and cysteine were spotted as standards. After drying, the TLC plate was treated with a ninhydrin solution (20 g ninhydrin in 600 ml ethanol) for amino-acid visualization. In this validated TLC set-up (*Yoshikawa et al., 2000*), cysteine and β-cyanoalanine can be identified both by color and relative mobility (Rf-value). β-cyanoalanine consistently displayed a blue-green color with a Rf value of around 0.8. In contrast, cysteine colored red and exhibited a consistent different Rf value.

The identification of β-cyanoalanine as a blue spot at 0.8 Rf was further confirmed by LC-MS analysis. After TLC separation, the zone around 0.8 Rf was scrapped from the TLC plate. After scrapping, the plate was colored as described above to confirm that the correct blue/green zone was collected. The collected silica was mixed with 200 μl ddH$_2$O, vortexed and centrifuged at 21000×*g* for 5 min. The supernatant was collected and directly analyzed using an Agilent 1100 Series LC-MSD with a diode array detector operating at 220 nm. The column oven was programmed at 35°C using a Phenomenex Luna 5u column (particle size) C18(2), 100A (pore size) with a column size of 250 × 3.0 mm. A gradient elution program driven by a quarternary pump was used at a flow rate of 0.5 ml/min (injection volume: 20 μl). The acetonitrile/water gradient used was 0–2 min (5% acetonitrile); 2–17 min (5–100% acetonitrile); 17–22 min (100% acetonitrile); 22–24.5 min (100–5% acetonitrile); 24.5–27 min (5% acetonitrile). The mass spectrometer was operated using the SCAN mode in the electrospray ionization mode. The analyzing sector contained a quadrupole analyzer and an electron multiplier detector.

## Acknowledgements

We thank S Bajda and P Zwaenepoel for performing PCR and LC-MS respectively and RM Clark, EJ Osborne and RT Greenhalgh for help and advice in constructing coverage plots.

---

## Additional information

### Funding

| Funder | Grant reference number | Author |
|---|---|---|
| Institute for the promotion of innovation by Science and Technology in Flanders (IWT) | SB/101451 | Nicky Wybouw |
| Fund for Scientific Research in Flanders (FWO) | | Wannes Dermauw, Thomas Van Leeuwen |
| Fund for Scientific Research in Flanders (FWO) | 3G061011 | Luc Tirry, Thomas Van Leeuwen |
| Ghent Special Research Fund | 01J13711 | Luc Tirry, Thomas Van Leeuwen |
| This work was partially supported by the government of Canada through Genome Canada and the Ontario Genomics Institute | OGI-046 | Miodrag Grbić |
| Fund for Scientific Research in Flanders (FWO) | 3G009312 | Thomas Van Leeuwen |

The funders had no role in study design, data collection and interpretation, or the decision to submit the work for publication.

## Author contributions
NW, Conception and design, Acquisition of data, Analysis and interpretation of data, Drafting and revising the article; WD, Acquisition of data, Analysis and interpretation of data, Revising the article; LT, MG, Revising the article, Contributed unpublished essential data or reagents; CS, Analysis and iRnterpretation of data, Contributed unpublished essential data or reagents; RF, Analysis and interpretation of data, Drafting and revising the article; TVL, Conception and design, Analysis and interpretation of data, Drafting and revising the article

## Additional files

### Supplementary files
• Supplementary file 1. Differentially expressed genes in the London-CYANO strain compared to the London strain.

• Supplementary file 2. Genome portals of arthropods consulted in order to retrieve potential homologues of Tu-CAS in addition to NCBI nr/nt databases.

• Supplementary file 3. List of the pyridoxal-5'-phosphate-dependent protein sequences used in phylogenetic tree construction.

• Supplementary file 4. List of primers used in PCR in *T. urticae* and *T. evansi*.

• Supplementary file 5. Coding sequences of *Tu-CAS*.

### Major dataset
The following dataset was generated:

| Author(s) | Year | Dataset title | Dataset ID and/or URL | Database, license, and accessibility information |
|---|---|---|---|---|
| Wybouw N, Dermauw W, Van Leeuwen T | 2013 | Genome wide gene-expression analysis of the spider mite *Tetranychus urticae* after long term host transfer from acyanogenic *Phaseolus vulgaris* cv. 'Prelude' bean plants to cyanogenic *Phaseolus lunatus* cv. '8078' bean plants | http://www.ncbi.nlm.nih.gov/geo/query/acc.cgi?acc=GSE50162 | Publicly available at NCBI GEO. |

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
