## [Decision Letter]

Thank you for sending your work entitled “A gene horizontally transferred from bacteria protects arthropods from host plant cyanide poisoning” for consideration at *eLife*. Your article has been favorably evaluated by a Senior editor and 3 reviewers, one of whom is a member of our Board of Reviewing Editors, and one of whom, Søren Bak, has agreed to reveal his identity.

The Reviewing editor and the other reviewers discussed their comments before reaching this decision, and the Reviewing editor has assembled the following comments to help you prepare a revised submission:

Your paper describes new evidence that supports a scenario of horizontal transfer of a gene encoding CAS/CYS activity from bacteria to *T. urticae*. The horizontally transferred gene has a proposed role in *T. uritcae's* detoxification of cyanide exposure from host plants. In general the external reviewers and the Reviewing editor found much that they liked about this work and recognized its novelty and relevance for a broad scientific audience. The paper appears to be important for our general understanding of the evolution of certain arthropods to colonize host plants that contain cyanogenic glucosides, and might lead to new genomics-enabled discoveries in other systems of plant-arthropod interactions. The paper is well written and easy to follow. The reviewers and the Reviewing editor found the paper convincing in its documentation of a CAS/CYS gene of likely bacterial origin integrated into the *T. urticae* genome. The biochemical characterization of the encoded *T. urticae* CAS/CYS enzyme also appears to be solid, although some additional data are requested to substantiate conclusions of product formation and enzyme kinetic properties (please see below). The reviewers and the Reviewing editor commented on a lack of discussion (or lack of data) to explain the phylogenetic pattern of a limited appearance of the CYS/CAS gene in *T. urticae* and only a relatively small number of Lepidoptera as presented in Figure 2. At the least, the Discussion of the paper needs to be substantially revised with a plausible explanation for this observation (please see below). However, if this important question regarding the phylogeny cannot be resolved with a new Discussion, additional data mining or generation of additional supporting data would be required to fill this critical gap in an otherwise interesting story.

Major issues to address:

1) To allow the reader to assess enzyme kinetic properties, the K_M_ and k_cat_ values must be shown in Table 1. V_max_ values shown in one of the figures are not very informative.

2) The text says that products of CAS activity, beta-cyanoalanine, were identified by TLC and LC-MS. The authors refer to Figure 9 for these results. However, Figure 9 only shows TLC data, which may not be sufficiently informative. To support statements of product identification, the LC-MS results must be shown for enzyme products and the authentic standard.

3) While the results appear to be generally sound, the reviewers and the Reviewing editor did not find a convincing explanation of the phylogenetic distribution of the described horizontally transferred *CAS/CYS* gene in *T. urticae* and only a few Lepidoptera. This point requires a better discussion and potentially additional data to support or explain the phylogenic pattern of distribution of the *CAS/CYS* gene.

The phylogenies in Figure 2 suggest that the horizontally transferred gene in the mite and the insects identified share a common ancestor. This is not discussed sufficiently in the text. The fact that this gene is apparently only found in 18 arthropods (of all of those with substantial sequences in public databases) would suggest either a loss of such a gene in most arthropods lineages or more than one independent horizontal gene transfer event in *T. urticae* and the Lepidoptera.

Although the authors indicate they looked at several insect species for this gene, they do not describe the extent of their search. Did they examine the genomic data available outside of Lepidoptera (e.g., in Coleoptera, Hymenoptera, and Diptera)? Between insects and arachnids, a common ancestor to this gene might suggest that it was introduced prior to the divergence of insects and arachnids, and therefore may be present (or vestiges thereof) in other insect species. Are there any? If not, does this suggest two separate horizontal gene transfer events? This in silico analysis, and an analysis of the predicted transfer event date based on divergence of the orthologous sequences, would add additional insight into this manuscript.

After addressing the above questions, the authors should discuss the alternative scenarios of a single vs. multiple horizontal gene transfer event(s) to explain the phylogeny described in Figure 2 and make a well-supported conclusion to this point.

---

## [Author Response]

*1) To allow the reader to assess enzyme kinetic properties, the K*_*M*_
*and K*_*cat*_
*values must be shown in*
Table 1*. V*_*max*_
*values shown in one of the figures are not very informative*.

The main objective of the enzyme kinetic data was to determine which of the two reactions (cysteine synthesis – CYS and β-cyanoalanine synthesis – CAS) was favored by the Tu-CAS enzyme. For this, a comparison of the specificity constants (k_cat_/K_M_) (a.k.a. second order rate constant) for each reaction was needed. The ratio of specificity constants (k_cat_/K_M_ of CAS activity over k_cat_/K_M_ of CYS activity) strongly favored CAS activity over CYS activity. As k_cat_ is directly proportional to V_max_ (k_cat_ = V_max_/[E]), the ratio is the same whether V_max_ or k_cat_ is used. The values given for the ratio of specificity constants are therefore identical and correct whether they are calculated from V_max_/K_M_ or k_cat_/K_M._ As the enzyme purity is not known precisely and this would introduce a systematic error, we calculated the V_max_/K_M_ ratio for the mite enzyme, as it cancels out the enzyme concentration in the equation. We have updated Table 1, so it now also includes the K_M_ and activity values.

The paragraph describing the kinetic data has now been amended in the Results section “Biochemical characterization of Tu-CAS”.

*2) The text says that products of CAS activity, beta-cyanoalanine, were identified by TLC and LC-MS. The authors refer to Figure 9 for these results. However, Figure 9 only shows TLC data, which may not be sufficiently informative. To support statements of product identification, the LC-MS results must be shown for enzyme products and the authentic standard*.

We have measured CAS activity using robust and well validated methods previously used for the bacterial and plant enzymes ([17]; Warrilow and Hawkesford 2000; [50]; Hatzfeld et al. 2000). These indirect methods are based on the colorimetric determination of sulfide (38). More directly, we have also shown the enzyme- and time-dependent formation of β-cyanoalanine after separation of reaction mixtures on TLC. This is especially powerful, as it allows the identification of β-cyanoalanine both by Rf value, and by a specific and strong color shift after treatment with a ninhydrin solution. The cyano group of β-cyanoalanine is responsible for a unique deep blue color (as also clear from the standard, see Figure 7, panel A, BCA standard). Finally, we scrapped the enzymatically produced β-cyanoalanine after separation of reaction mixtures from silica plates, and analyzed it by LC-MS. We showed that the elution time on LC and the characteristic ion of m/z = 113 which is [M-H]^-^ of the β-cyanoalanine standard are also found in the enzymatic product. We have added figure panels to illustrate the LC-MS data as requested (Figure 7 panel B1 and B2).

*3) While the results appear to be generally sound, the reviewers and the Reviewing editor did not find a convincing explanation of the phylogenetic distribution of the described horizontally transferred CAS/CYS gene in T. urticae and only a few Lepidoptera. This point requires a better discussion and potentially additional data to support or explain the phylogenic pattern of distribution of the CAS/CYS gene*.

*The phylogenies in*
Figure 2
*suggest that the horizontally transferred gene in the mite and the insects identified share a common ancestor. This is not discussed sufficiently in the text. The fact that this gene is apparently only found in 18 arthropods (of all of those with substantial sequences in public databases) would suggest either a loss of such a gene in most arthropods lineages or more than one independent horizontal gene transfer event in T. urticae and the Lepidoptera*.

*Although the authors indicate they looked at several insect species for this gene, they do not describe the extent of their search. Did they examine the genomic data available outside of Lepidoptera (e.g., in Coleoptera, Hymenoptera, and Diptera)? Between insects and arachnids, a common ancestor to this gene might suggest that it was introduced prior to the divergence of insects and arachnids, and therefore may be present (or vestiges thereof) in other insect species. Are there any? If not, does this suggest two separate horizontal gene transfer events? This in silico analysis, and an analysis of the predicted transfer event date based on divergence of the orthologous sequences, would add additional insight into this manuscript*.

*After addressing the above questions, the authors should discuss the alternative scenarios of a single vs. multiple horizontal gene transfer event(s) to explain the phylogeny described in*
Figure 2
*and make a well-supported conclusion to this point*.

By a first series of BLAST-searches in NCBI databases (nr/nt), Tu-CAS homologues could not be found outside Lepidoptera even though the genomic data of many other arthropods are present in NCBI. Nevertheless, we have meticulously screened a large number of genome portals directly, and an exhaustive list of databases searched is now included as [Supplementary-material SD2-data], without revealing any sequence that was not previously detected.

In order to better describe the extent of our search, we altered the first paragraph of the Results section “Phylogeny of Tu-CAS and evidence for a bacterial origin by horizontal transfer”.

We thank the referees and editors for pointing out that the phylogeny should be better discussed in the manuscript. We have now gone as far as the data allow without gratuitous speculation, and we have added the section beginning “A homologous lateral gene transfer has also occurred in Lepidoptera” to the first paragraph of the Discussion.

We also updated Figure 2 now showing also the phylogenetic position of the CAS enzymes of related tetranychid mite species *Tetranychus evansi* and *Panonychus citri*.